# A Membrane with Strong Resistance to Organic and Biological Fouling Using Graphene Oxide and D-Tyrosine as Modifiers

**DOI:** 10.3390/membranes12050486

**Published:** 2022-04-29

**Authors:** Jiarui Guo, Yan Zhang, Fenghua Chen, Yuman Chai

**Affiliations:** 1Key Laboratory of Water Quality Science and Water Environment Recovery Engineering, Beijing University of Technology, Beijing 100124, China; guojiarui@emails.bjut.edu.cn; 2Faculty of Architecture, Civil and Transportation Engineering, Beijing University of Technology, Beijing 100124, China; chenfh@cnpe.cc (F.C.); chaiyuman@163.com (Y.C.)

**Keywords:** membrane fouling, mixed matrix membranes, D-amino acid, graphene oxide (GO), biofouling control

## Abstract

Membrane fouling markedly influences the service life and performance of the membrane during the using process. Herein, hydrophilic polyvinylidene fluoride (PVDF) nanocomposite (P-GO-DAA) membranes with antifouling and anti-biofouling characteristics were fabricated by employing graphene oxide (GO) and different concentrations of D-Tyrosine. The structural properties of the prepared nanocomposite membranes as well as pure PVDF membranes were characterized using FTIR, XPS, SEM, AFM, and contact angle analysis. It was found that the introduction of GO fillers made an excellent antifouling performance compared to pure PVDF indicated by the pure water flux, flux recovery rate, and rejection rate during ultrafiltration experiments as a result of the formation of the hydrophilic and more porous membrane. In particular, the nanocomposite membranes showed an increased flux of 305.27 L/(m^2^·h) and the rejection of 93.40% for the mixed pollutants solution (including Bull Serum Albumin, Sodium Alginate, and Humic Acid). Besides, the outstanding anti-biofouling activity was shown by the P-GO-DAA membrane with the properties of D-Tyrosine for inhibiting biofilm formation during the bacterial adhesion experiments. Furthermore, the adhesion ratio of bacteria on the membrane was 26.64% of the P-GO-DAA membrane compared to 84.22% of pure PVDF. These results were confirmed by CLSM.

## 1. Introduction

Ultrafiltration (UF) membranes have been considered as a promising strategy for wastewater treatment [1,2,3]. Nevertheless, a massive quantity of foulants in the environment such as heavy metal ions, suspended solids, natural organic matters, microorganisms, and others, is likely to accumulate on the membrane surface leading to the organic fouling and biofouling of the membrane [4,5].

Nanomaterials are commonly added as a modifier to the membrane matrix or surface to enhance the permeability and the antifouling property of the membranes [6,7,8,9]. Among them, GO and functionalized GO have been recognized as attractive materials to prepare nanocomposite membranes owing to their high hydrophilic, porous structure, the abundance of oxygen-containing functional groups, and strong mechanical properties [10,11,12]. Zwitterionic modified GO sheets were used to prepare PVDF nanocomposite membranes for improving the hydrophilic property, which reduced the water contact angle to 65.1° [13]. A nanocomposite forward osmosis (FO) membrane was synthesized by using polyvinylpyrrolidone (PVP) modified GO to enhance the membrane permeability, with a water flux of 33.2 L/(m^2^·h), which was 3.3 times higher compared with the pristine FO membranes [14]. Khan et al. fabricated a membrane blended hybrid nanosheets by using the aid of the functional groups of GO, which induced covalent organic frameworks (COFs) to GO, exhibiting the highest water flux of 226.3 L/(m^2^·h) [15].

GO contributes significantly to the improvement of membrane resistance to organic foulants in the nanocomposite membranes. Recently, a novel polysulfide (PSF) membrane used vanillin-modified GO nanosheets as pore formers was reported with a rejection rate of 99%, and the membrane showed a flux recovery rate of 88.55% when filtering BSA as a pollutant [16]. Similarly, polyethersulfone (PES)/sulfonated polysulfone (SPSf) nanocomposite membranes were prepared by using a very low GO content of 0.012 wt% as a modifier, with a high flux recovery rate of 92.4% for BSA filtration [17]. Wang et al. fabricated a self-assembly polyacrylonitrile membrane with GO to improve the antifouling capability of the membrane, achieving the flux recovery rate up to 91.2% for HA [18]. GO makes the membrane more resistant to organic fouling, but it has a limited effect on the biofouling of the membrane surface.

Biofouling is usually inevitable due to the inherent properties of microorganisms as well as the accumulation of biologically active organisms along with extracellular polymeric substances (EPS) on the membrane surface [19]. That brings up the tricky problem which is, even if 99.9% of them are eliminated, there are still enough cells remaining that could continue to grow by using biodegradable substances in the water [20]. Consequently, biofouling is inescapable unless it is sustained in sterile surroundings or no nutrients are present at all [20,21,22].

Presently, the widely used strategy to deal with biofouling is inducing Ag, Cu, and heavy metal materials to the matrix of the membrane, to achieve their effect through the lethal action on microorganisms. Sun et al. used GO-Ag nanoparticles to develop the biofouling resistant membranes, which found the presence of GO-Ag nanoparticles on the membrane led to an inactivation of 86% *E. coli* after contact with the membrane [23]. However, the excessive release of biocidal materials is unfriendly to non-target microorganisms and even deteriorates the surroundings [24,25]. Hence, natural materials as eco-friendly substitutions for antimicrobial agents have become a huge focus, such as chitosan [26], D-amino acid (DAA) [27], and pancreatic enzymes [28].

D-Tyrosine (a typical DAA), a newly discovered green substance, was demonstrated to mitigate bacterial fouling by prompting self-disintegrating of biofilm effectively at extremely low concentrations [29,30,31,32,33]. Auto-inducers (AIs) are known as signal molecules that elicit quorum sensing (QS) of microorganisms to coordinate their collective behavior including biofilm formation [34]. A series of works in the available literature has confirmed that DAA could prevent the synthesis of bacteria by altering their cell wall composition, leading to the loss of the ability of bacteria to secrete AIs normally, which triggers the disassembly of biofilms [35,36,37]. Based on the explorations above, D-Tyrosine has been applied in membrane technology to suppress bacterial adhesion to the membrane. Yu et al. developed an anti-biofouling membrane by incorporating D-Tyrosine onto a membrane using zeolite nanoparticles that inhibited biofilm formation without inactivating the bacteria [36]. Guo et al. prepared DAA-modified PVDF nanocomposite membranes supported by PDA and halloysite nanotubes (HNTs), and a stable anti-biofouling over 10 days was obtained for the prepared membrane [38].

The DAA-modified membranes usually introduce DAA through adhesion on the surface of the membrane [38,39,40]. Khan et al. modified an anti-biofouling membrane by using alginate dialdehyde (ADA) to graft D-Tyrosine on the membrane surface, and this membrane could achieve nearly 80% bacterial inhibition [39]. Jiang et al. modified the membrane by adhering D-Tyrosine through polydopamine (PDA) on the membrane surface, and the bacterial attachment rate of the membrane decreased by about 10% after modification [40]. The membranes mentioned above exhibited superior anti-biofouling performance, but the water permeability of the membranes may be partly influenced.

Herein, in this study, GO incorporated DAA-modified PVDF membranes (P-GO-DAA) with strong resistance to organic and biological fouling were prepared by blending GO nanosheets into the PVDF matrix, and followed by the introduction of D-Tyrosine through hydrogen bonding. The characteristics and performances of the membranes were analyzed through FTIR, XPS, SEM, AFM, and contact angles. The antifouling activity of the prepared membranes was investigated by filtrating multiple typical organic foulants including bull serum albumin (BSA), sodium alginate (SA), and humic acid (HA). The anti-biofouling activity was evaluated through the bacterial adhesion tests and cyclic filtration tests using *E. coli* as the model microorganism.

## 2. Materials and Methods

### 2.1. Materials and Chemicals

All reagents and chemicals are of analytical grade and used as received. To prepare membranes, Polyvinylidene fluoride (PVDF), polyvinylpyrrolidone (PVP), and N, N-dimethylacetamide (DMAC) were all obtained from Shanghai Maclin Biochemical Technology Company. GO (Sheet diameter is 0.5~5 μm and thickness is 0.8~1.2 nm) used as membrane matrix modifier was purchased from Nanjing Xianfeng Nanomaterials Technology Company. D-Tyrosine as an antibacterial agent was supplied by Perfemiker. BSA (Shanghai Maclin Biochemical Technology Company, Shanghai, China), SA, and HA (Tianjin Fuchen Chemical Reagent Company, Tianjin, China) were used as typical organic foulants. *E. coli* used as a biological foulant was purchased from Guangzhou Strain Preservation Center. LB broth and LIVE/DEAD^®^ BacLight Bacterial Viability Kits were supplied by Thermo Fisher Technology (Shanghai, China) Company. DI water was purified by a Millipore Direct-Q3 in this paper.

### 2.2. Preparation of PVDF, P-GO, P-GO-DAA Membranes

PVDF and P-GO membranes were prepared via the non-solvent induced phase separation (NIPS) technique and the percentages of all of the components in the membrane casting solutions are shown in Appendix A. For the preparation of the PVDF membrane, a homogeneous cast solution was prepared by dissolving PVP as the pore-forming agent and PVDF in DMAC with the aid of a magnetic stirrer at 45 °C for 12 h. After the solution placed in an oven at 60 °C for more than 6 h to release the trapped gas bubbles, it was cast using a 250 μm scraper on clean glass plates at room temperature. The cast membrane was left for 30 s to partially evaporate the solvent and immersed in DI water that served as a non-solvent bath. The resulting membrane was washed with DI water and soaked in DI water for later use when it was detached from the glass surface within tens of seconds. Similarly, the P-GO membrane was fabricated by the same process but GO was dispersed in DMAC in advance by sonication for 30 min at room temperature.

The P-GO-DAA membranes were modified by separately immersing the P-GO membranes in D -tyrosine solutions (DI water as solvent) at different concentrations of 50, 100 and 150 mg/L at 45 °C for 24 h. for modification. The samples were referred to as P-GO-50, P-GO-100 and P-GO-150, respectively.

The synthesis schematic was shown in Figure 1.

### 2.3. Characteristics of Membranes

The morphology of membrane surface and cross-section was probed by scanning electron microscope (SEM; SU8020, Hitachi, Tokyo, Japan). The cross-section samples were pretreatment by brittle fracturing with the aid of liquid nitrogen and the surface was covered with a thin gold sputtering layer employing an ion sputter device (Buehler, Lake Bluff, IL, USA) prior to the SEM analysis. Atomic force microscopy (AFM; Dimension Icon, Bruker AXS, Karlsruhe, Germany) was implemented using a three-direction closed-loop scanner to obverse the roughness of the membrane surface.

The element composition and functional groups were identified through X-ray photoelectron spectroscopy (XPS; ESCALAB 250Xi, Thermo, Waltham, MA, USA) and Fourier transform infrared (FTIR; V70, Bruker, Karlsruhe, Germany) spectroscopy with a diamond attenuated total reflection (ATR) accessory.

The hydrophilicity was investigated via static contact angle conducted by goniometer (OCA50, Dataphysics, Filderstadt, Germany) to reflect the wettability properties of the membrane surface.

The average pore size of membranes was analyzed with an absorption and desorption instrument (Autosorb iQ, Quantachrome Instruments, Boynton Beach, FL, USA). The porosity (ε) of the membrane was measured by the gravimetric method as reported before and calculated as [41]:(1)$$\varepsilon =\frac{\frac{w_{w}-w_{d}}{\rho_{w}}}{\frac{w_{w}-w_{d}}{\rho_{w}}+\frac{w_{d}}{\rho_{p}}}\times 100\%$$
where the $w_{w}$ and $w_{d}$ denote the weight (g) of the wet and dry membrane, respectively, $\rho_{w}$ and $\rho_{p}$ represent the densities (g/cm^3^) of DI water and PVDF, respectively.

### 2.4. The Permeability of the Membranes

The performance of the pristine and modified membranes was systemically assessed by a lab-scale dead-end filtration unit (Figure 2) at a stable trans-membrane pressure of 0.1 MPa supplied by a nitrogen cylinder. The membranes were initially pre-pressed with DI water at the pressure of 0.2 MPa for 20 min before testing. For the DI water permeation test, the pure water through the membrane was collected every 5 min for 30 min and the flux $J_{w}$ was calculated by Equation (2):


(2)
$$J_{w}=\frac{V}{A\times \Delta_{t}}$$


Here *V* denotes the volume of permeated water (L); *A* denotes the effective filtration membrane area in m^2^ and Δt represents the filtration time in hours (h).

### 2.5. Separate Performance and Antifouling Activity

To determine the separation and antifouling capacity of the membranes, the ultrafiltration experiments were performed again with several kinds of feed water solutions instead of DI water, and the contents and concentrations of these solutions are shown in Table 1.

The pure water flux (*J*_0_) was recorded continuously for 30 min after the pressure stabilized, and the permeate (*J_F_*) passing through the membrane was also recorded for 30 min when the DI water was replaced with the polluted feed water. Later, the coupons were backwashed by DI water completely prior to the measure of pure water flux (*J_R_*) of the backwashed membranes. From these experiments, five parameters were calculated: *rejection rate*, flux recover rate (*FRR*, %), reversible fouling rate (*R_r_*), irreversible fouling rate (*R_ir_*), and total fouling rate (*R_t_*). *Rejection rate* and *FRR* were calculated as follows:(3)$$rejection\;rate=\left(1-\frac{C_{P}}{C_{F}}\right)\times 100\%$$
(4)$$FRR=\frac{J_{R}}{J_{0}}\times 100\%$$
where the *C_F_* and *C_P_* denote the organic content measured with a TOC meter (Vario TOC, Element, Munich, Germany) of feed water and permeate water (mg/L) collected from filtration experiments.

While *R_r_*, *R_ir_* and *R_t_* were calculated using Equations (5)–(7), respectively:(5)$$R_{r}=\frac{J_{R}-J_{F}}{J_{0}}\times 100\%$$
(6)$$R_{ir}=\frac{J_{0}-J_{R}}{J_{0}}\times 100\%$$
(7)$$R_{t}=R_{r}+R_{ir}=\left(1-\frac{J_{F}}{J_{0}}\right)\times 100\%$$

### 2.6. Anti-Biofouling Activity

The antibacterial efficiency of the membranes was evaluated by employing *E. coli* as the model microorganism. The test bacterium was cultivated in a Luria-Bertani (LB) liquid medium using a modified method reported before [42]. The cultivated suspension with *E. coli* 10 mL was poured into a conical flask containing a membrane sample sterilized and then incubated at 37 °C for 5 d. The concentration of the *E. coli* suspension was measured with a UV spectrophotometer at wavelength 650 nm, and the membranes were cut into coupons and then flushed with DI water for further testing.

In order to visualize the bacterial adhesion and distinguish the activity of bacteria on the membrane surface, the bacteria on the membranes were colored by the LIVE/DEAD Bacterial Viability Kit and the stained membrane samples were observed under the confocal laser scanning electron microscopy (CLSM, LSM800, Zeiss, Jena, Germany).

### 2.7. Characterization of Comprehensive Fouling and Biofouling Resistance Activity

To determine the antifouling and anti-biofouling activities of the membranes, the flux, concentration of TOC and the adhesion of *E. coli* on the membrane surface were measured with the synthetic heavily polluted sewage containing *E. coli* suspension by the three cycles of ultrafiltration experiments. Subsequently, the adhesion and growth of bacteria on the membrane surface were investigated by the CLSM and SEM to characterize the membrane’s resistance to biological foulants.

## 3. Results and Discussion

### 3.1. Characterization of PVDF Membranes

The FTIR spectra were performed to investigate the functional groups of the prepared membranes. Figure 3 shows the FTIR spectra for the pristine PVDF, P-GO, and P-GO-DAA membranes from 500 cm^−1^ to 4000 cm^−1^. The infrared absorption peaks with corresponding chemical bonds are shown in Appendix A. Compared with the PVDF membrane, a higher level of O-H and C=O for the carboxyl group observed in the P-GO membrane at the peak of 3385 cm^−1^ and 1720 cm^−1^, respectively, indicated the successful introduction of GO in the P-GO membrane. The increase in intensity at 1413 cm^−1^ and 860 cm^−1^ for the P-GO-DAA membrane corresponding to the deformation and stretching of N-H and C-N [43], confirmed the existence of DAA on the P-GO-DAA membrane. After the incorporation of DAA with the P-GO membrane, the peaks at 1615 cm^−1^ and 1720 cm^−1^ for the P-GO membrane that ascribed to the bending vibration of carboxyl and hydroxyl groups [44], shifted to a broad vibration band at 1653 cm^−1^ for the P-GO-DAA membrane. Likewise, the peaks of 1320 cm^−1^ that reflected O-H bond of phenol present in D-Tyrosine shifted to 1280 cm^−1^ for the P-GO-DAA membrane. These changes in peak position may be due to intermolecular hydrogen bonding between D-Tyrosine and GO [44,45]. To further clarify the connection between GO and DAA, the mixture of DAA and GO in solution was tested and the same changes in peak positions were observed in the GO-DAA by the FTIR spectra (Appendix A). It was these peaks shifted that speculated the presence of intermolecular hydrogen bonding between D-Tyrosine and GO. Besides, the peak at 1037 cm^−1^ represents C-O-C vibration of GO shifted to 1069 cm^−1^ in the GO-DAA, which speculated the possible intermolecular hydrogen bonding between the epoxy group of GO and D-Tyrosine. The suggestions for interaction are shown in Appendix A.

XPS provided further confirmation of the transformations of the functional groups and elemental compositions on the surface of the membranes. As shown in Figure 4, the C1s narrow sweep spectrum for different membranes could be distinguished into C-C, C-F, C=C and C-O/C-N species, which peaks located at 284.8, 289.3, 283.6, and 286.2 eV, respectively. The increased peak area of C=C compared with the PVDF membrane was testified to the presence of GO in the membrane matrix. The slight enhancement of the peak area of C-O/C-N was also probably owing to the limited distribution of GO on the membrane surface. For the P-GO-DAA membrane, the increased proportion of O and N for the P-GO-DAA membrane (Appendix A) compared to the P-GO membrane confirmed that the P-GO membrane was successfully modified with DAA. The increased peak area of C-O/C-N was also gained by incorporating the DAA since DAA possessed amino, hydroxyl and carboxyl groups.

The surface and cross-sectional images of the prepared membranes were taken by SEM and presented in Figure 5. The SEM images showed the PVDF and nanocomposite membranes had a porous structure on the surface as well as an asymmetric cross-section structure consisting of a compact surface layer and a sponge-like porous structure beneath. The pristine PVDF membrane possessed fewer pores with a minimal number of disconnected voids possibly due to its high hydrophobicity [46]. After GO was induced to the membrane matrix, a much more porous surface with relatively interconnected large pores in the bulk and a thinner surface layer (Appendix A) were observed in the P-GO membrane. For the DAA-modified membranes, a denser and compressed layer on top of the membrane was observed which seemed to be increased with the concentration of D-Tyrosine in the membrane. This may be related to the molecular weight and conformation of the D-Tyrosine immobilized on the membrane surface, and this kind of morphological feature was supposed to promote membrane separation performance [47].

The roughness has a relatively great influence on the water permeability and antifouling property of the membrane. High roughness would lead to easier deposition of pollutions on the membrane surface, and very low roughness might diminish the tendency of water molecules through the membrane [48,49]. The surface roughness of each fabricated membrane was determined using AFM images (Figure 6). The P-GO membrane displayed the lowest roughness, while the P-GO-DAA membrane possessed higher roughness than the P-GO membrane and lower roughness than the PVDF membrane. The addition of hydrophilic GO could facilitate the formation of a smoother surface during the membrane casting process when using the DI water as the non-solvent bath. The presence of D-Tyrosine changed the morphology of the nanocomposite membrane surface in a way leading to higher surface roughness that depended on the number of D-Tyrosine on the membrane surface. As expected, the P-GO-DAA membrane had a more appropriate surface roughness in contrast with other membranes, which would better balance the permeability and antifouling ability of the membrane.

The average pore size and porosity were analyzed and the results were listed in Table 2. It was found that the average pore size of the P-GO membrane reduced after the addition of GO, but the porosity of the membranes had improved inversely. This may be since the existence of GO in the P-GO membrane might disrupt the continuity of the membrane matrix resulting in more voids in the membrane. With the introduction of D-Tyrosine, the average pore size and porosity of the P-GO-DAA membranes showed a very minor difference in comparison with the P-GO membrane, which could be presumed that the addition of D-Tyrosine did not have a noticeable impact on the membrane pores.

To evaluate the wettability of the membranes, water contact angles analyses (Figure 7) of membranes were carried out by the static contact angle test. The highest contact angle was exhibited by the pure PVDF with 82.4°, while the P-GO membrane was 72.27°. The contact angle of the P-GO-DAA membranes decreased even more to 67.77°, 64.67°, and 60.27° for the P-GO-50, P-GO-100, and P-GO-150 membranes, respectively. The presence of numerous hydrophilic groups (such as the hydroxyl and carboxyl groups) around the boundaries of GO nanosheets endowed the PVDF membrane with higher hydrophilicity [18]. There is a positive correlation between the hydrophilic of the membrane and the DAA content that may not only be attributed to the increased polar functional groups on the membrane surface, but also to the increased surface roughness.

### 3.2. Pure Water Permeability of the Membranes

The permeation flux of the membranes plays a crucial role in the membrane application. The filtrations with pure water were performed, and the results were depicted in Figure 8. The addition of GO to the PVDF casting solution could increase the pure water flux due to the improved porous network and surface hydrophilicity of the membrane. 0.2 wt% of GO in the casting solution was enough to raise the pure water flux from 138.33 L/(m^2^·h) (for pure PVDF) to 282.08 L/(m^2^·h). A more increase in permeation flux was observed for the P-GO-DAA membranes, with the maximum reaching 305.27 L/(m^2^·h) for the P-GO-100 membrane, which was more than twice compared with the pure PVDF. This higher permeability was probably a combined result that the improved hydrophilicity of the membrane surface by adding GO and DAA to the membrane, as well as a higher surface roughness produced by the modification using DAA, allowing water molecules to pass through the membrane more easily [46]. Nevertheless, a small decrease in the pure water flux for the P-GO-150 membrane could be attributed to a small amount of pore narrowing that occurred due to the relatively high concentration of DAA added. This phenomenon revealed that the water permeation would decline beyond a certain DAA content in the membranes. Overall, the contribution of GO to the increased flux of the membranes was more, and similar kinds of results that nanoparticles contributed more to the water flux of membranes compared to DAA have been reported [26].

### 3.3. Separate Performance of the Membranes

In order to study the separate performance of the prepared membranes, filtrations with different aqueous solutions containing single and mixed foulants were performed (Figure 8). For single foulants, it was seen from Figure 8a that the P-GO membrane had good *rejection rates* with 67.25%, 78.87%, and 83.75% compared with the pure PVDF with 44.98%, 47.70% and 71.05% for BSA, SA, and HA, respectively. This increase suggested the separate capacity of the membrane could be contributed to the reduction of the average pore size caused by the blending of GO. After the addition of D-Tyrosine, the highest rejections were observed in the P-GO-DAA membranes for BSA, SA, and HA around 84%, 90% and 95%, respectively, and this increase was due to a dense layer created by the immobilized of DAA on the P-GO membrane surface. The rejection capacity of the membrane for different pollutants had a great relationship with the membrane pore size and the size of pollutant molecules [50]. The membrane rejection for BSA gained the lowest result could be owing to the smaller molecule size of BSA used in this experiment than SA and HA.

As expected, the separation performance of membranes was manifested to be better while employing the solutions comprising complex foulants as feed water during the filtration tests (Figure 8b). The *rejection rates* of the P-GO-150 membrane for Bi, Tri, and AS reached 82.53%, 93.40%, and 86.82%, respectively, which were superior to those of the pure PVDF. This result might be strongly related to a more complex interaction between pollutants leading to the formation of larger agglomerates, so the *rejection rate* of the membranes for the triple foulants solution was higher than the binary foulants solution. The *rejection rate* of membrane dropped for actual sewage which was likely due to the presence of smaller molecule foulants in actual sewage [51,52]. These favorable results indicated that the P-GO-DAA membranes possess an enhanced separate performance for various organic foulants. The difference in these *rejection rates* among the DAA-modified membranes was not significant, which could be ascribed to the addition of D-Tyrosine having little effect on the membrane pores (Table 2), leading to a similar rejection capacity. Similar results were found by Khan et al. [39].

### 3.4. Antifouling Performance of the Membranes

Flux recovery rate is one of the most crucial measures to reveal the antifouling properties of membranes. The *FRR* of the prepared membranes was tested through filtrations with single or compound foulants and the results were shown in Figure 9. The P-GO-DAA membranes exhibited better flux recovery performance for single foulants, especially the P-GO-150 membrane, having the highest *FRR* for BSA, SA and HA with 76.61%, 87.87%, and 61.65%, respectively. The *FRR* of the P-GO-150 membrane for Bi, Tri, and AS reached 74.58%, 88.72%, and 86.67%, respectively, which were about 28%, 38%, and 39% higher than the P-GO membrane, and about 64%, 64%, and 70% higher than the PVDF membrane. Except for the more hydrophilic membrane surface induced by adding GO and DAA, this improvement could also be a result that DAA reduced the interaction between membrane and foulants leading to less adsorption of foulants by the membrane [17,53,54]. FRR of the P-GO-DAA membranes for all pollutions increased with increasing DAA content, which denoted the antifouling ability might have a positively correlated with the content of DAA in the membrane.

Meanwhile, *R_r_* and *R_ir_* were also employed as evaluation indexes of membrane antifouling activity (Figure 9), and the accumulation of irreversible fouling would directly descend the performance and service life of the membrane. For all DAA-modified membranes, the percentage of irreversible fouling by various pollutants was obviously reduced. The *R_r_* for AS of the P-GO-150 membrane was 13.85%, which decreased markedly compared with that of PVDF (84.31%) and P-GO (52.33%) membrane. It could be speculated that various interactions between foulants and foulants or foulants and membrane of complex multicomponent systems retarded the fouling instead. As most foulants were hydrophobic, the interaction between the foulants and the membrane was weakened with the assistance of hydrophilic GO and DAA, which greatly improved the membrane’s ability to resist organic fouling. A higher proportion of reversible fouling was obtained with increasing DAA content. This might be attributed to the fact that the higher the roughness, the easier it is to remove the foulants attached to the membrane surface, resulting in less fouling remaining in the membrane.

### 3.5. Anti-Biofouling Performance of the Membranes

To investigate the effect of DAA on bacterial fouling, a five-day static adhesion experiment was conducted using *E. coli* as the model bacteria and the CLSM images (Figure 10 and Figure 11) of the membranes were taken during the experiment. It can be observed that the P-GO-DAA membranes possess strong antibacterial activity as the less bacterial adhesion extent on the membrane surfaces. On the first day, the pure PVDF membrane had already exhibited worse resistance to biofouling. In comparison almost no bacteria on the P-GO-DAA membrane surface. As time passed, the bacteria adhering to the origin membrane surface showed a rapid growth trend. Until the fifth day, a large number of bacteria adhered to the surface of the PVDF membrane (as the stronger the fluorescence in the images, the greater the number of bacteria), as well as the live and dead bacteria on the surface of the membrane co-formed the biofilm. On the contrary, only a small area of the modified membrane surface was fouled and almost no biofilm formation during the period. It was apparent that the number of bacteria on the membrane surface decreased after D-Tyrosine grafted. Hence, the P-GO-DAA membrane exhibited better performance to biofouling, possibly due to the fact that the presence of D-Tyrosine not only prevented bacteria from adhering to the membrane but also promoted the detachment of biofilm from the membrane surface by influencing the secretion of QS signals of the bacteria [36].

### 3.6. Antifouling and Anti-Biofouling Performance of the Membranes through Cycle Test

The antifouling and anti-biofouling performance of the membranes were studied by the cyclic filtration experiments using a heavily polluted solution containing BSA, SA, HA, and *E. coli* (Figure 12). The flux (Figure 12a) of the modified membranes decreased slowly during filtration using the polluted solution as the feed water, while the PVDF membrane was already reduced dramatically after the first cycle, even down to 2 L/(m^2^·h). On the contrary, the P-GO-100 membrane exhibited a higher flux than the origin membrane and maintained the flux of 158 L/(m^2^·h) after three cycles, even higher than the initial flux of the PVDF membrane (138.5 L/(m^2^·h)). The *FRR* (Figure 12b) of the P-GO-DAA membranes was relatively higher during the whole cycles, especially for the P-GO-100 and P-GO-150 membranes. Among the prepared membranes, the *rejection rate* of the P-GO-DAA membrane was maintained at a higher level during the filtration process. The difference in *rejection rates* between the DAA-modified membranes was not significant, which was similar to the results observed before in Figure 9. It was seen from Appendix A that the average pore size of the pure PVDF dropped considerably after the test, indicating the serious pore blockage of the origin membrane, while the pores of the DAA-modified membrane did not change much.

The outstanding anti-biofouling capacity of the modified membranes was also showed in Figure 12c. In contrast with the P-GO-DAA membranes, the PVDF membrane and the P-GO membrane seemed to have no bacterial inhibition with the bacteria adhering ratio after three cycles reaching 84.22% and 80.98%, respectively. The bacteria adhering ratio of the P-GO-100 membrane was only 26.64%, about 60% and 56% lower than the PVDF and P-GO membranes, respectively, indicating an excellent resistance to bacterial adhesion on the D-Tyrosine modified membranes. It was demonstrated that the addition of D-Tyrosine played a greater role in the antibacterial property. The CLSM images of the membranes (the two membranes circled in red in Figure 12c) after the experiment (Figure 13) also demonstrated the conclusions above mentioned. Both live and dead bacteria that adhered to the pure PVDF were significantly more than the modified membranes. Fewer bacteria adhered to the surface of the modified membrane. It was also indicated that probably due to the presence of DAA with the inhibiting effect of QS signals diminishing the bacterial attachment on the membrane surface.

The improved antifouling and anti-biofouling performances of the P-GO-DAA membranes were verified by SEM images (Figure 14). The surface roughness of the DAA-modified membrane did not have much change after the experiment comparing to the origin membrane.

To sum up, the addition of GO and D-Tyrosine to the membrane could weaken the interaction between the foulants and membrane surface due to the improved hydrophilic of the membrane, so that pollutants on the membrane surface are more easily carried away by the shear force of water. On the other hand, the addition of DAA could effectively inhibit bacterial adhesion and induce biofilm self-decomposition.

## 4. Conclusions

In this study, a P-GO-DAA membrane with strong resistance to organic and biological fouling was developed via the phase inversion technique by using GO and D-Tyrosine as modifiers. The FTIR results suggested that DAA might be introduced into the membrane by forming hydrogen bonds with GO. The addition of DAA to membrane formed a dense layer on the membrane surface observed from SEM. The porous structure and high hydrophilic nature of GO led to enhanced pore structure and surface hydrophilicity of the nanocomposite membranes, which were verified by the increasing porosity and decreasing water contact angle. By adding GO combined with DAA, the P-GO-DAA membrane exhibited a further improvement in pure water flux up to 305.27 L/(m^2^·h) that was more than two folds of the PVDF membrane, and a more hydrophilic surface was confirmed by the decline in water contact angle compared with the PVDF membrane. The modified membranes displayed better separate ability contrast with the pure PVDF. The P-GO-DAA membrane had the best separation performance with a rejection rate of 95% especially for HA. The antifouling ability of the P-GO-DAA membrane became stronger evidenced by its significantly increased FRR for various foulants, especially for actual sewage by 70% compared to the origin membrane. For the anti-biofouling activity of the membranes, the addition of DAA played a key role with the results of 60% and 56% less surface bacterial adhesion than of the PVDF and P-GO membranes, respectively, and almost no bacteria adhered to the P-GO-DAA membrane also observed by CLSM. It was supposed that DAA could diminish the interaction between the membrane and bacteria by impacting the secretion of QS signals. Consequently, it could be concluded that the GO blended DAA-modified nanocomposite membranes possess a synergistic effect on the antifouling and anti-biofouling activity of the membrane.

## Figures and Tables

**Figure 1 membranes-12-00486-f001:**
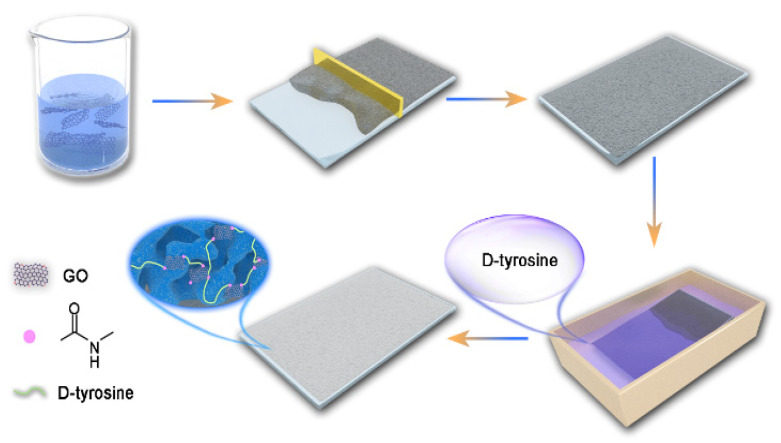
Synthesis schematic of membranes.

**Figure 2 membranes-12-00486-f002:**
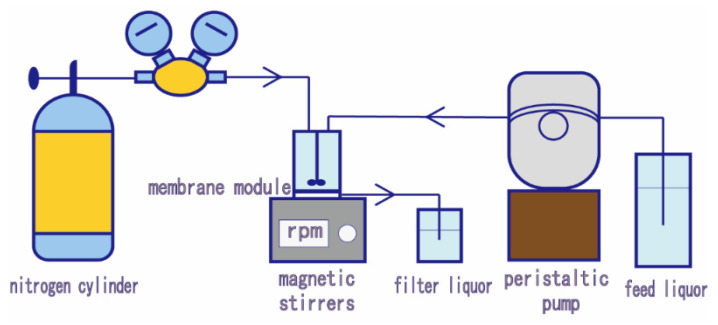
Lab-scale dead-end filtration unit.

**Figure 3 membranes-12-00486-f003:**
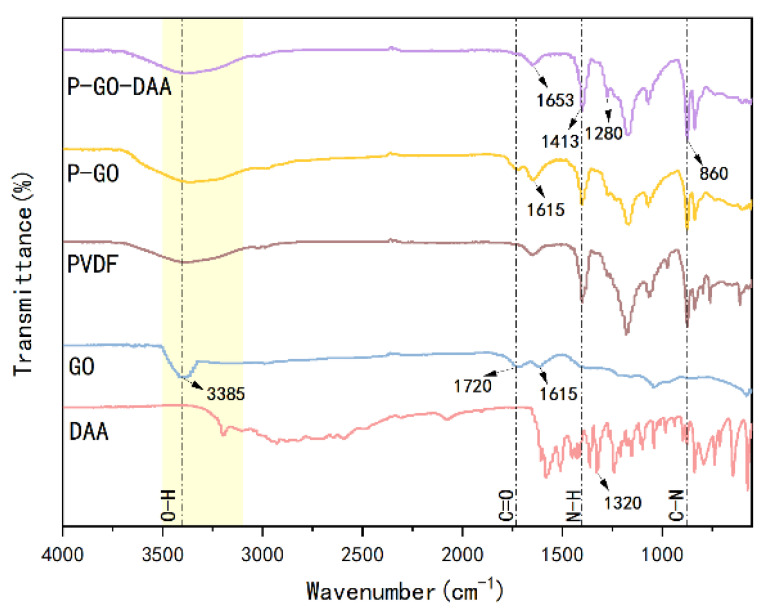
FTIR spectra of DAA, GO, PVDF, P-GO, and P-GO-DAA membranes.

**Figure 4 membranes-12-00486-f004:**
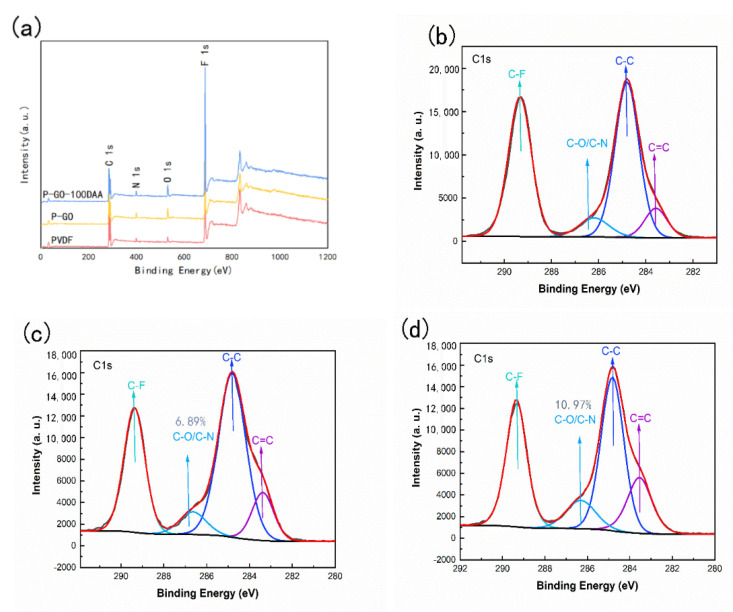
XPS characterization of nanocomposites. (**a**) XPS spectra of PVDF, P-GO, and P-GO-DAA membranes; (**b**) C1s XPS spectra of PVDF; (**c**) C1s XPS spectra of P-GO; (**d**) C1s XPS spectra of P-GO-DAA membranes.

**Figure 5 membranes-12-00486-f005:**
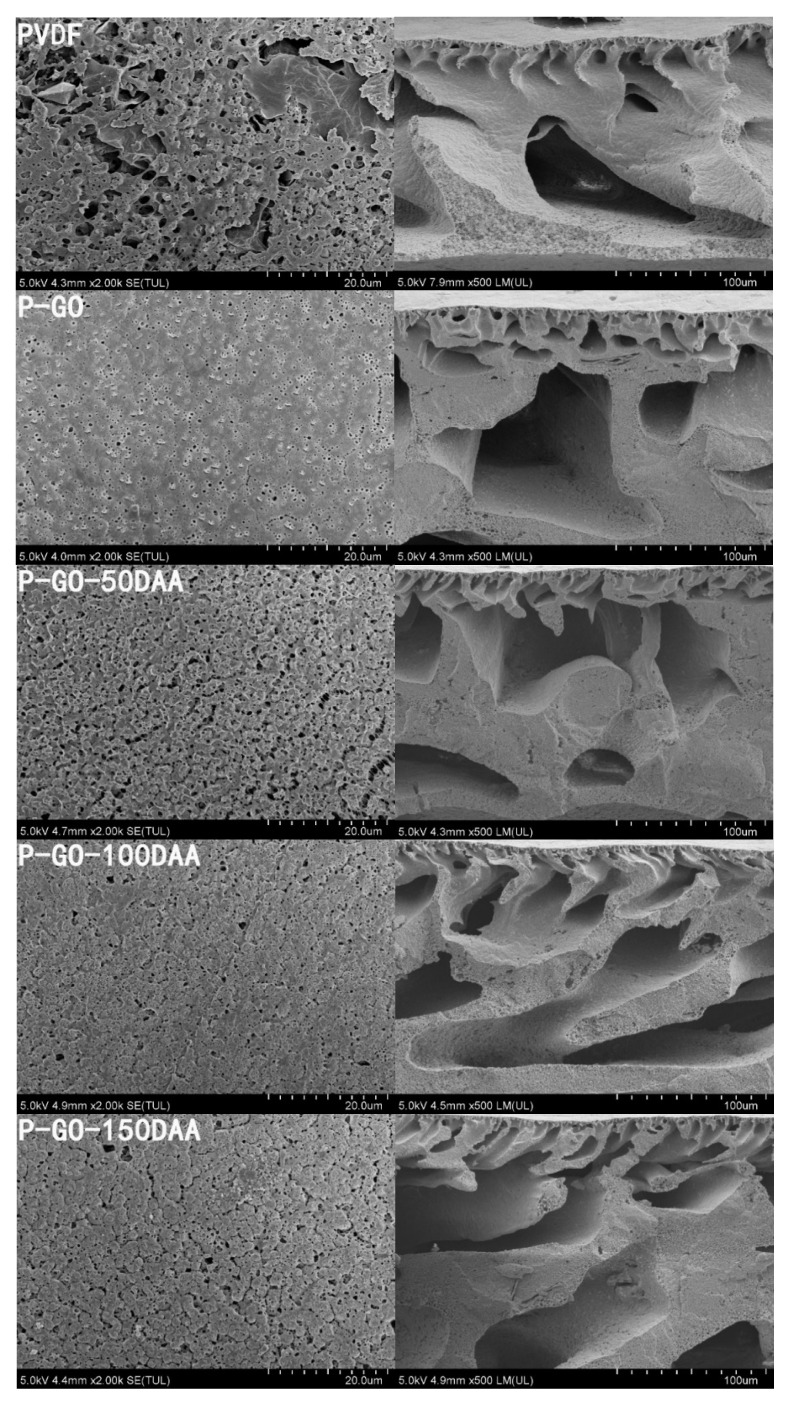
SEM images of the surface and cross-sectional morphologies of different membranes.

**Figure 6 membranes-12-00486-f006:**
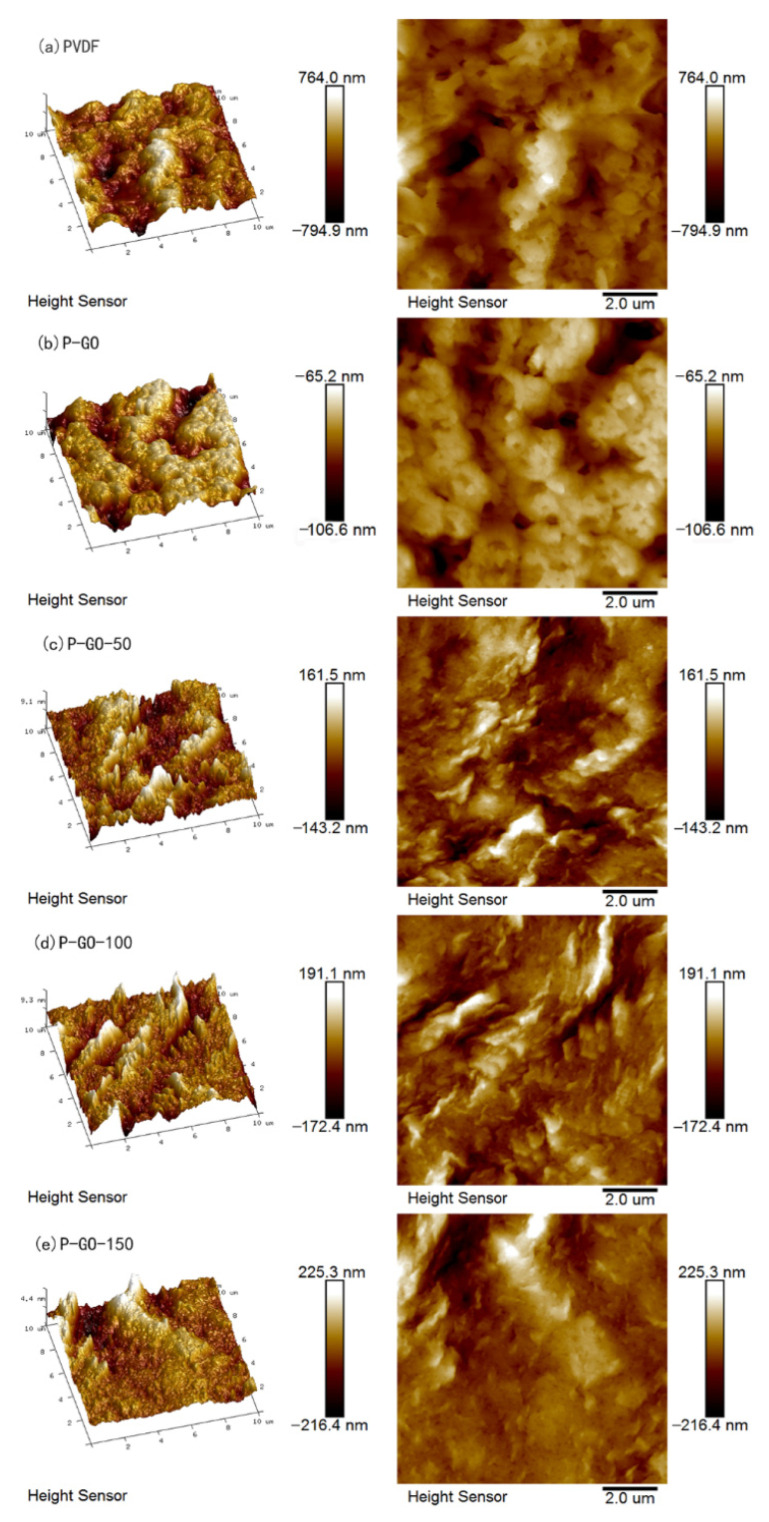
AFM images of various membranes.

**Figure 7 membranes-12-00486-f007:**
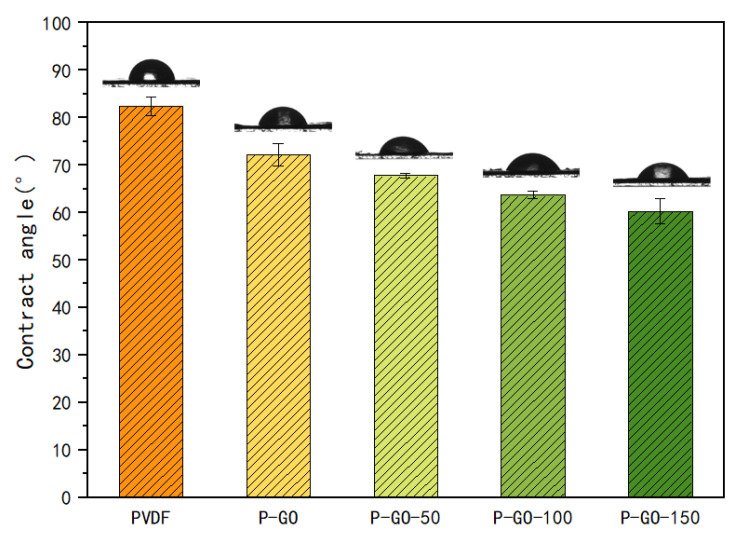
The contact angle of PVDF, P-GO, P-GO-50, P-GO-100, P-GO-150 membrane. The CA data and images were obtained by dropping 2.0 μL DI water on five different locations for each membrane surface sample at room temperature.

**Figure 8 membranes-12-00486-f008:**
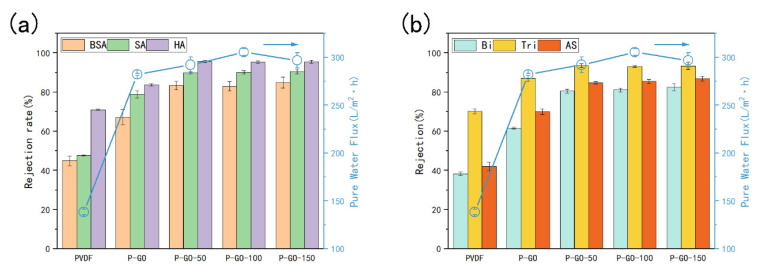
Pure water permeability and *rejection rate* for various foulants of pristine and functioned PVDF membranes: (**a**) *rejection rate* for BSA, SA, and HA; (**b**) *rejection rate* for Bi (binary foulants: BSA and SA), Tri (triple foulants: BSA, SA, and HA), and AS (Actual sewage: effluent of secondary sedimentation tank).

**Figure 9 membranes-12-00486-f009:**
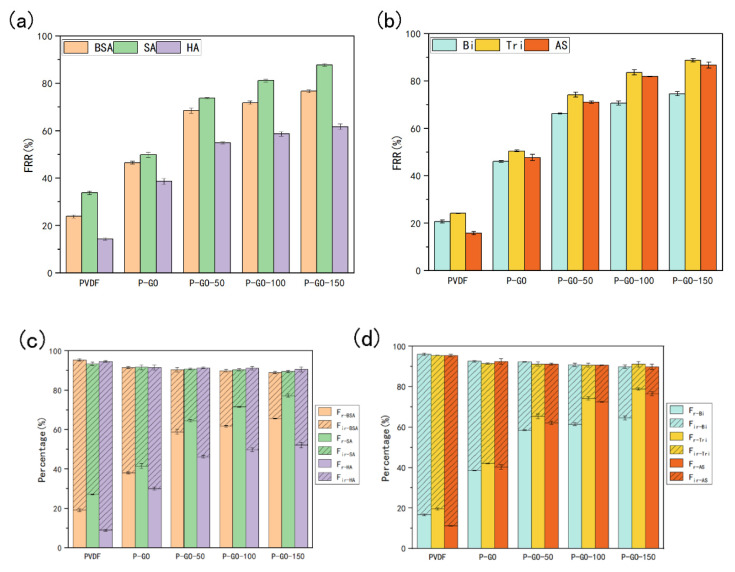
Flux recovery ratio (*FRR*), reversible flux decline ratio (*R_r_*) and the irreversible flux decline ratio (*R_ir_*) for different foulants during filtration experiments of original and prepared membranes. (**a**,**b**) *FRR* of the membranes; (**c**,**d**) *R_r_* and *R_ir_* of the membranes.

**Figure 10 membranes-12-00486-f010:**
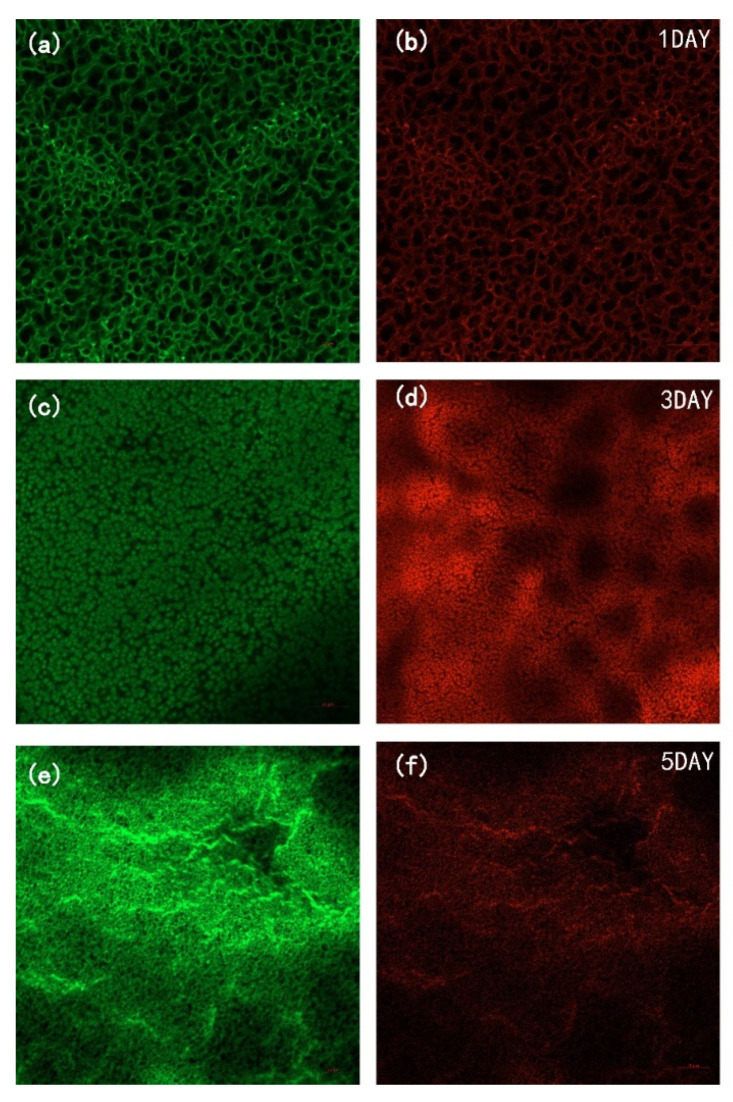
CLSM images of PVDF membrane during five days of bacterial culture cycle. (**a**,**c**,**e**) Live bacteria on PVDF membrane; (**b**,**d**,**f**) dead bacteria on PVDF membrane.

**Figure 11 membranes-12-00486-f011:**
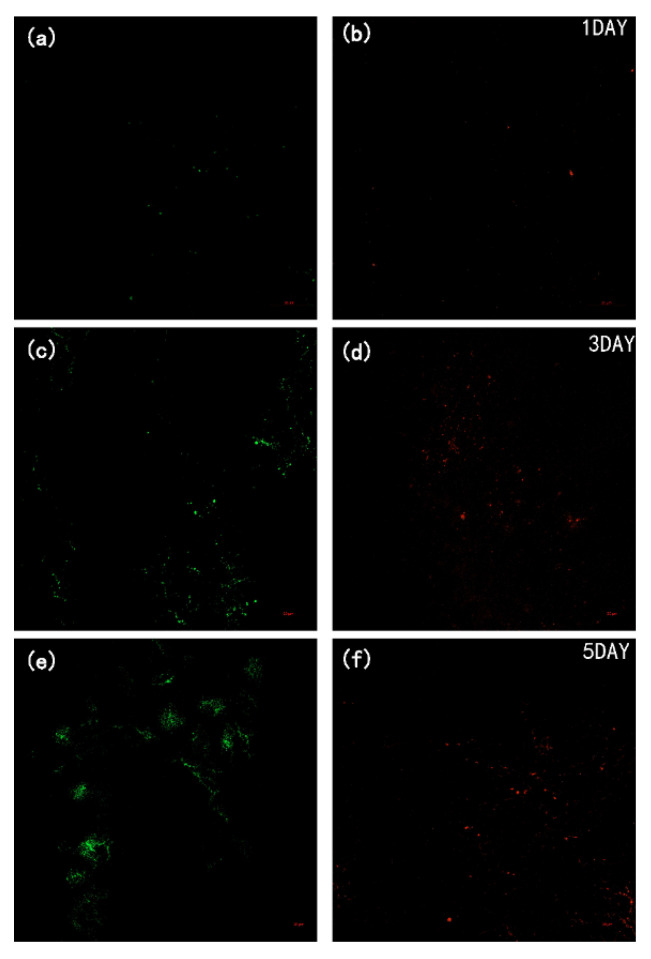
CLSM images of P-GO-DAA membrane during five days of bacterial culture cycle. (**a**,**c**,**e**) live bacteria on PVDF membrane; (**b**,**d**,**f**) dead bacteria on PVDF membrane.

**Figure 12 membranes-12-00486-f012:**
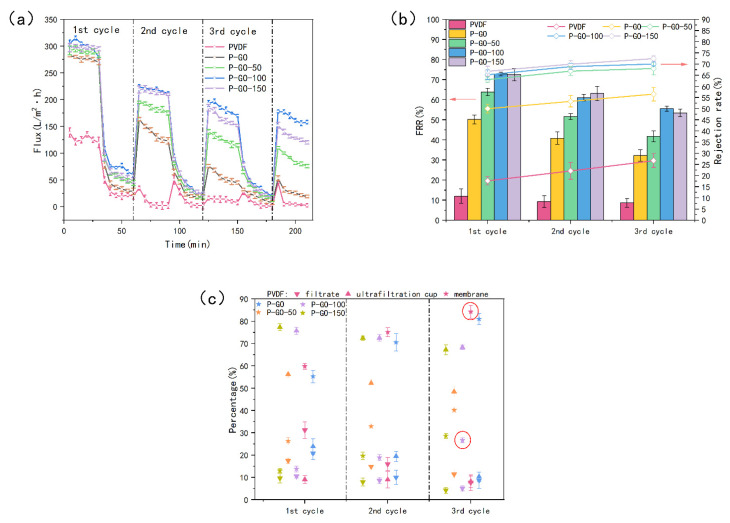
The results of three cycles ultrafiltration experiment. (**a**) The membrane flux change; (**b**) flux recovery rate and *rejection rate* of membranes; (**c**) the bacterial distribution in the test.

**Figure 13 membranes-12-00486-f013:**
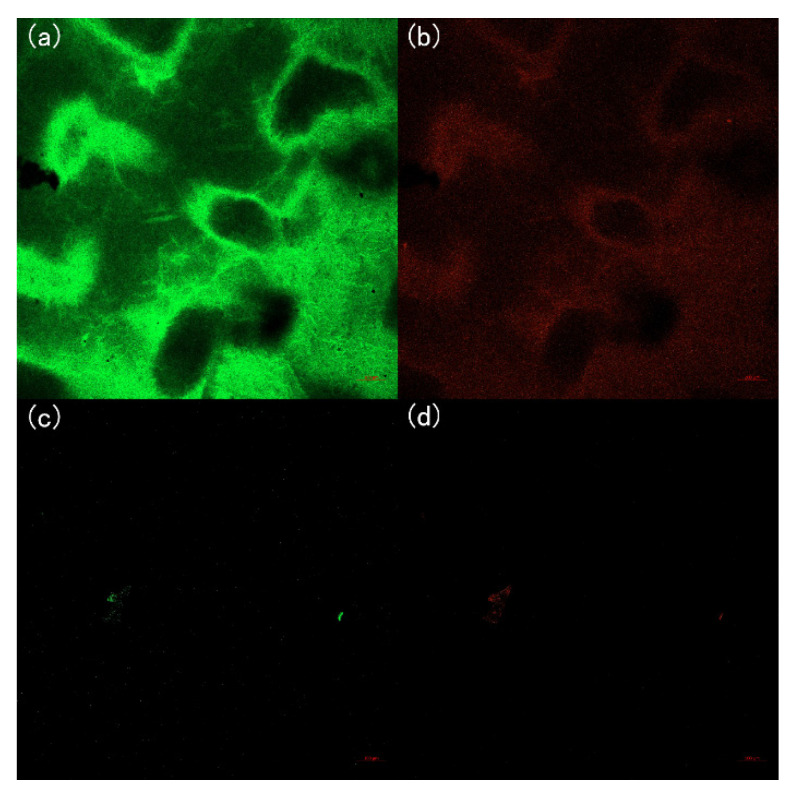
CLSM images of the membranes after the three cycles ultrafiltration experiment. (**a**) Live bacteria on the origin membrane; (**b**) dead bacteria on the origin membrane; (**c**) live bacteria on the P-GO-DAA membrane; (**d**) dead bacteria on the P-GO-DAA membrane.

**Figure 14 membranes-12-00486-f014:**
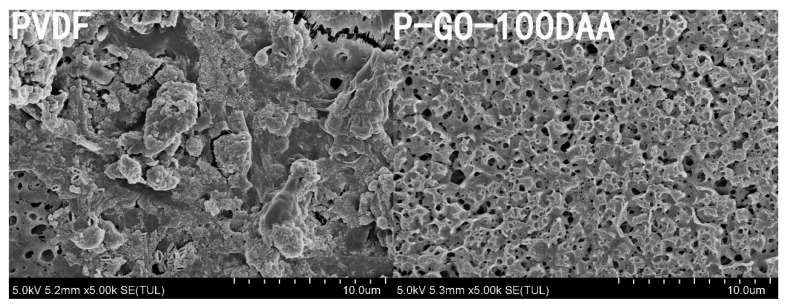
SEM images of the membranes after the three cycles ultrafiltration experiment.

**Table 1 membranes-12-00486-t001:** Composition of pollutant solutions.

Name of the Solution	Contents and Concentrations
BSA (Bull Serum Albumin)	BSA, 100 mg/L
SA (Sodium alga Acid)	SA, 50 mg/L
HA (Humic Acid)	HA, 50 mg/L
Bi (Binary pollutants)	BSA, 100 mg/L SA, 50 mg/L
Tri (Triple pollutants)	BSA, 100 mg/L SA, 50 mg/L HA, 50 mg/L
AS (Actual Sewage)	Effluent from a secondary sedimentation tank in a sewage plant in Beijing, TOC: 182 mg/L
Synthetic pollutants solution	BSA, 100 mg/L SA, 100 mg/L HA, 50 mg/L *E. coli*, 3.78 × 10^7^ CFU

**Table 2 membranes-12-00486-t002:** Average pore size and porosity of membranes.

Membrane	Average Pore Size (nm)	Porosity (%)
PVDF	16.9 (±2.1)	74.5 (±1.4)
P-GO	16.2 (±1.6)	79.1 (±1.8)
P-GO-50	15.8 (±1.3)	80.3 (±2.5)
P-GO-100	15.7 (±1.8)	81.0 (±2.3)
P-GO-150	14.4 (±2.0)	79.6 (±2.7)

## Data Availability

Not applicable.

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
