# Peer review of "A Membrane with Strong Resistance to Organic and Biological Fouling Using Graphene Oxide and D-Tyrosine as Modifiers"

_membranes, 2022, doi:10.3390/membranes12050486_

Round 1

Reviewer 1 Report

The purpose of the article is to present modification of PVDF ultrafiltration membranes with Graphene Oxide and subsequently with D-Tyrosine towards creating anti-fouling membranes for water treatment. Several experimental methods were employed to evaluate how the GO and DAA enhance the anti-fouling properties of the membranes. There are several key issues with the experimental design, presentation, interpretation of results and some control experiments. There is no mention of how stable the GO matrix and the DAA on the surface is. There is no mention of the surface coverage of the DAA molecules but almost all antibacterial properties have been attributed to the hydrophilicity imparted by the carboxyl and amine groups of DAA. 

  1. Abstract is very generic. The summary of results and the list of methods needs to be mentioned in the abstract. The abstract should clearly describe what the manuscript will be about.
  2. Line 43 is a repetition of lines 36-37.
  3. Lines 132-136 seem to be from the manuscript template. Please edit the whole manuscript. 
  4. Line 154, What was the solvent used for making D-Tyrosine solutions? I am assuming DI water but it is not mentioned anywhere.
  5. Line 157-158, please format correctly. 
  6. Line 188, "neat and fabricated" is not correct descriptions. Please change to "pristine and modified"
  7. Line 191, replace "percolate" with "permeate".
  8. Line 206, correct "gauge" spelling.
  9. Line 237, please provide the instrument specifications, brand, model for CLSM.
  10. Line 253, There is no N-H in PVDF. Please provide the correct interpretation of the IR spectrum. The peak at 860cm-1 is not prominent.
  11. Line 256, Why does the C-N stretching weaken. Is the C-N part of the GO?
  12. The explanation of the FTIR is not thorough. There are no schematic representations of the chemistry or the chemical reactions being referred to in the FTIR spectra. It would be very difficult for everyone to comprehend the chemistry being discussed.
  13. Figure 3. GO Spectrum is not well resolved and not clear at all
  14. Lines 269-270, 

    Can the authors highlight this significant rise in C-OH? it is not clear from the figure.

    Simple deposition of D-tyrosine would also lead to higher C-O concentrations. This is not ncessarily a proof of crosslinking of DAA and GO. There is not schematic representation of the chemistry anywhere.

  15. Table 5. What are the standard deviations? Without the errors, it is difficult to conclude whether these values are similar or different .
  16. Figure 8, the differences in the permeabilities of the GO and DAA modified membranes, are very insignificant. The explanations given for the differences within the GO-DAA (50, 10 and 150) membrane is not convincing. 
  17. Figure 10 is wrongly captioned. 
  18. Figure 11, during the cyclic experiments, have authors tried to find if there is any loss of DAA? why is the flux declining?
  19. Why were the CLSM and FRR studies carried out only with PVDF and P-GO-100DAA? What about the other 3 membranes?
  20. Finally, the PVDF coating with just DAA has to be studied as the other control. As per the manuscript, there is no clear understanding of what GO is contributing and what DAA is contributing ,The chemical structure on the surface has not been validated. 

Author Response

Dear Reviewer 1:

        Thank you for your letter and for the reviewer’s comments concerning our manuscript entitled “A Novel membrane with strong resistance to organic and bio-logical fouling using graphene oxide and D-tyrosine as modi-fiers” (ID: membranes-1626362). Those comments are all valuable and very helpful for revising and improving our paper, as well as the important guiding significance to our researches. We have studied comments carefully and have made correction which we hope meet with approval. Revised portion are marked in red in the paper. The main corrections in the paper and the responds to the reviewer’s comments are in the word document.

Reviewer 2 Report

The manuscript by Guo et al. presented the MMM preparation based on PVDF/PVP with GO and treated with D-Tyrosine for UF application and presented enhanced separation performances and improved anti-biofouling ability.

First and foremost, the manuscript is hard to read (primarily the abstract and introduction sections).

  1. The article needs serious language revision. The authors must simplify the word selections and sentences (bombastic and complicated words are not needed).
  2. Reduce/shorten the sentences. There is no need to combine 2/3/4 sentences, when it can be much more straightforward, shorter sentences. Wrong grammar and punctuation make reading the manuscript a lot harder.

Besides that, listed below are my suggestions:

  1. In the introduction, please specify the details (polymer, performances, etc.) when discussing other works – the ‘improved a lot, showed outstanding, ultra-high rejection’… means nothing here.
  2. The authors mentioned that the addition of nanoparticles in MMM is mainly to improve hydrophilicity/break the limitation of the membrane’s hydrophobicity. The nanoparticles contribute more that than. Please re-discuss.
  3. Please define BSA, SA and HA when the terms are first used.
  4. Line 132-136. What are these?
  5. How long is the membrane immersion, Line 154.
  6. Section 3.1.1 – the discussion needs to be revised. The authors emphasise that D-Tyrosine is grafted to GO, which I think is not entirely accurate. The GO is already in the polymer matrix when the D-Tyrosine-soaking (related to comment 5) and the degree of grafting would highly depend on the morphology of the polymer and soak duration. Also, GO may have ‘reacted’ with PDVP and PVP during the dope blending.
  7. Maybe tables 3 and 4 are not needed. Move to SI or incorporate to the text. If you’re moving it to the SI – please consider moving Table 1 too.
  8. Pore size and porosity discussion need to be revised.
    1. The porosity increase caused by GO (at only 0.2% loading) is not convincing. Most probably wrong. Please consider other explanations, i.e., polymer matrix disruption etc.
    2. How is the main cause of porosity change related to the DAA-GO interaction – when these two are less likely to contribute! GO is in the deep-polymer matrix (with maybe some – out that 0.2% at the near-surface).
  9. Section 3.1.2
    1. Please measure the surface layer and indicate it in the SEM images.
    2. The discussion on Line 289-292. What are the authors trying to explain?
    3. Line 306-308 means nothing here. Please link it to AFM findings (graphs included but not discussed). And for AFM, please discuss appropriately, i.e. surface morphology 3D information.
    4. Line 321-324 – this explanation is not correct. Please refer to the lotus effect.
    5. Line 324-326 – not needed. It is more complex than that, so better exclude it.
  10. Section 3.2.1 – Too wordy, and please discuss better. And when it’s needed, please include the values and not just ‘smaller, higher, etc.).
  11. Figure 11 to Section 3.4.
  12. Figure 10 to section 3.3. and please include border, numbering (a,b…) and captioned appropriately.
  13. Section 3.4.
    1. The same comment – please include value when mentioning higher, smaller, lower etc. It makes more sense and makes the discussion more meaningful.
    2. Line 478-479 – SEM does not prove any antifouling ability. The statement is not appropriate. It would be better if the authors could link its smoothness (in addition to its chemical characteristic) and relate to the anti-fouling behaviour.
    3. Even better if you combine SEM before and after the cycles and measure the surface pore size.
  14. Small formatting mistakes:
    1. Line #52 – use proper subscript.

Overall, the manuscript needs to be revised, and attention should be given to the discussion.

Author Response

Dear Reviewer 2:

     Thank you for your letter and for the reviewer’s comments concerning our manuscript entitled “A Novel membrane with strong resistance to organic and bio-logical fouling using graphene oxide and D-tyrosine as modi-fiers” (ID: membranes-1626362). Those comments are all valuable and very helpful for revising and improving our paper, as well as the important guiding significance to our researches. We have studied comments carefully and have made correction which we hope meet with approval. Revised portion are marked in red in the paper.

     Thank you very much for your question about our language, we have made some targeted changes. We have simplified the statements and changed some words and grammatical errors to meet your expectations.

     The main corrections in the paper and the responds to the reviewer’s comments are as in the word document.

Reviewer 3 Report

While the paper is good at presenting and discussing the results, a lot of aspects could be improved for this paper.

  1. Recommend the authors proofread the manuscript. Authors tend to use unnecessary long sentences but shorter sentences could be easier to follow. Also the use of prepositions has massive problems. Please make changes.
  2. The format of the paper should be checked. For instance, the word "sterilized" on the 2nd line of page 4 used a different font. The tables should follow three-line table format but it is not consistent in the manuscript. 
  3. In section 3.1.2, there was only one short sentence discussing AFM images. It is not sufficient. 
  4. The authors have filtration and recovery experiments, and anti-bacteria tests in the draft, however, there was no long-time filtration study to prove the dynamic anti-fouling effectiveness of the membranes. Please add an experiment for stronger evidence. 

Author Response

Dear Editors and Reviewer 3:

     Thank you for your letter and for the reviewer’s comments concerning our manuscript entitled “A Novel membrane with strong resistance to organic and bio-logical fouling using graphene oxide and D-tyrosine as modi-fiers” (ID: membranes-1626362). Those comments are all valuable and very helpful for revising and improving our paper, as well as the important guiding significance to our researches. We have studied comments carefully and have made correction which we hope meet with approval. Revised portion are marked in red in the paper.

     Thank you very much for your question about our language, we have made some targeted changes. We have simplified the statements and changed some words and grammatical errors to meet your expectations.

     The main corrections in the paper and the responds to the reviewer’s comments are as in the word document.

Round 2

Reviewer 1 Report

I thank the authors for attempting to correct the errors and clarifying the issues raised in the previous review. However, overall the work lacks novelty. The conclusions drawn a purely based on the hydrophillic property supposedly imparted by D-tyrosine and it was not conclusively shown that the the D-tyrosine has a direct effect on the antifouling. Also, BSA rejection, flux and pure water permeability do not show any correlation with DAA content. Without a clear understanding of the DAA effect on the membrane properties and eventually biofouling performance, this work is not novel. 

Reviewer 2 Report

Thank you Guo et al for the revised manuscript. Here a few of my comments:

  1. Needs to be reread again and again. There are still many language errors, and please consider simplifying and using fewer words. Using accurate commas would help too. Wrong choice of words and expressions occur in many places.
  2. Line 38 - foulant instead of fouling. Line 50 functionalized instead of functioned. Please recheck the whole manuscript again. Line 235 - functional group. There are many others, so please recheck.
  3. Use correct spacing between the last word and citation brackets
  4. Pay attention to detail. Line 59 - ref 26 is about polysulfone and not polysulfide. Same with ref 19.
  5. Move Table 1 to the section it was first introduced.

Overall, the manuscript has improved a lot and I thank the authors for putting a lot of effort to improve it. My only technical concern is: how sure the authors are about the interaction between GO and tyrosine (I'm still not entirely convince) - my suggestion is to do a quick test where GO is reacted with tyrosine under the same soaking condition and check for FTIR. The findings can be added further into the SI. 

Reviewer 3 Report

While the manuscript is in a much better shape than the previous version, it is helpful if the authors can list a table of FTIR peak versus the wavelength for better understanding.

Also, please explain why use mean pore radius? Also, the mean pore size data did not show statistically different, though the authors state that "the average pore size of GO is smaller than the original membrane". Are the authors talking about average pore size or mean pore size? And if so, with the deviation, where is the statistic difference?

Round 3

Reviewer 1 Report

Thank you authors for making the changes and revising the manuscript.